# Dynamic Changes in Protein-Membrane Association for Regulating Photosynthetic Electron Transport

**DOI:** 10.3390/cells10051216

**Published:** 2021-05-16

**Authors:** Marine Messant, Anja Krieger-Liszkay, Ginga Shimakawa

**Affiliations:** 1Institute for Integrative Biology of the Cell (I2BC), CEA, CNRS, Université Paris-Saclay, CEDEX, 91198 Gif-sur-Yvette, France; marine.messant@cea.fr; 2Research Center for Solar Energy Chemistry, Osaka University, 1-3 Machikaneyama, Toyonaka, Osaka 560-8531, Japan; gshimakawa@kwansei.ac.jp; 3Department of Bioscience, School of Biological and Environmental Sciences, Kwansei-Gakuin University, 2-1 Gakuen, Sanda, Hyogo 669-1337, Japan

**Keywords:** photosynthesis, regulation, abiotic stress, membrane association, thylakoid membrane

## Abstract

Photosynthesis has to work efficiently in contrasting environments such as in shade and full sun. Rapid changes in light intensity and over-reduction of the photosynthetic electron transport chain cause production of reactive oxygen species, which can potentially damage the photosynthetic apparatus. Thus, to avoid such damage, photosynthetic electron transport is regulated on many levels, including light absorption in antenna, electron transfer reactions in the reaction centers, and consumption of ATP and NADPH in different metabolic pathways. Many regulatory mechanisms involve the movement of protein-pigment complexes within the thylakoid membrane. Furthermore, a certain number of chloroplast proteins exist in different oligomerization states, which temporally associate to the thylakoid membrane and modulate their activity. This review starts by giving a short overview of the lipid composition of the chloroplast membranes, followed by describing supercomplex formation in cyclic electron flow. Protein movements involved in the various mechanisms of non-photochemical quenching, including thermal dissipation, state transitions and the photosystem II damage–repair cycle are detailed. We highlight the importance of changes in the oligomerization state of VIPP and of the plastid terminal oxidase PTOX and discuss the factors that may be responsible for these changes. Photosynthesis-related protein movements and organization states of certain proteins all play a role in acclimation of the photosynthetic organism to the environment.

## 1. Introduction

The thylakoid membrane in chloroplasts is a complex three-dimensional structure that is morphologically highly dynamic, as seen by rearrangements of protein complexes under low light and high light conditions. The thylakoid membrane harbors the photosynthetic electron transport chain. It is known that several processes that regulate the efficiency of light harvesting and electron transport are linked to membrane dynamics. The electron transport chain is made out of the main protein complexes photosystem II (PSII), the site of water oxidation and oxygen release, the cytochrome *b*_6_*f* complex (Cyt *b*_6_*f*) and photosystem I (PSI), where ferredoxin is reduced. Reduced ferredoxin reduces NADP^+^ to NADPH, as catalyzed by the Ferredoxin NADP^+^ oxidoreductase (FNR). Both photosystems contain chlorophylls and carotenoids in their reaction centers, in the inner antenna and in the light harvesting complexes (LHC). During photosynthetic electron transport, a proton gradient (ΔpH) is established that is, together with an electrochemical gradient (ΔΨ), the driving force for the synthesis of ATP by the CF_0_CF_1_-ATP synthase. Regulation of light absorption efficiency of the pigment containing antenna systems and of photosynthetic electron transport is essential to allow the plant to acclimate rapidly to changes in light intensities, which can occur, for example, in sun flecks, when shade leaves or whole plants are suddenly exposed to full sun light. Regulation of photosynthetic electron transport is also needed to fulfil the energetic demands of the chloroplast/cell/organism by optimizing the ATP/NADPH ratio. This can be achieved by changing the activity of cyclic electron transport around PSI, a process requiring formation and dissolution of supercomplexes. Well-characterized processes involved in the distribution of the LHCs between the photosystems, the so-called state transitions, depend on movements of large protein-pigment complexes within the membrane. Another well-known process that involves protein movements within the thylakoid membrane is the PSII damage–repair cycle. Furthermore, soluble proteins attach reversibly to the thylakoid, enabling their involvement in metabolic reactions and regulatory mechanisms (Figure 1).

In this review, we first shortly describe the specific characteristics of the thylakoid membrane. Then, we focus on cyclic electron flow, alterations of the efficiency of light absorption and dissipation of excess energy as heat (non-photochemical quenching, NPQ), followed by a chapter on state transitions and the PSII damage–repair cycle. We will present other less well-known processes that are involved in the acclimation response of the photosynthetic apparatus and which depend on dynamic interactions with the thylakoid membrane (Figure 2). We focus mainly on processes taking place in eukaryotic photosynthetic organisms.

## 2. Membrane Composition

Chloroplast membranes have a very unique lipid composition with the galactolipids monogalactosyldiacylglycerol (MGDG), digalactosyldiacylglycerol (DGDG) and the glycolipid sulfoquinovosyl-diacylglycerol (SQDG) [1,2,3,4]. The external and the internal envelope membranes and the thylakoids have different lipid compositions. The outer envelope membrane is rather permeable, especially for ions, and is important for the connection between the chloroplast and the endoplasmic reticulum (ER), where protein translation principally takes place. It is composed of about 25% DGDG and 30% phosphatidylcholine, the main lipid of the ER membrane. On the contrary, the inner envelope membrane is particularly impermeable and serves as a barrier. The thylakoid membranes are mainly made of MGDG and DGDG (MGDG about 50%, DGDG about 30% of total lipid content). The lipids SQDG and phosphatidylglycerol (PG), the only major phospholipid in thylakoids, are negatively charged at neutral pH. Each represents about 10% of the total lipid content. The physicochemical properties of the head groups of the individual lipids are important for their behavior. MGDG and DGDG are neutral lipids with uncharged polar head groups. MGDG is not able to form a lipid bilayer in water because of its galactose head group [2,3]. In the thylakoid membranes, however, it forms a lipid bilayer thanks to the presence of a very high protein concentration. Compared with MGDG, DGDG has a bulkier headgroup with a cylindrical shape thanks to the presence of a second galactose residue allowing it to form a lamellar lipid bilayer. The glycolipids constitute the lipid matrix of the thylakoid membrane. As pointed out by Demé et al. [5] and Kobayashi et al. [6], the ratio of MGDG to DGDG is important for the phase behavior of the lipid bilayer, and thereby for the structure and stability of the thylakoid membrane. The negatively charged lipids SQDG and PG may play a role in the interaction with peripheral membrane proteins. 

Thylakoid membranes form distinct structures: the so-called grana stacks and stroma lamellae that differ in their protein composition. The outer layer of the grana stacks and the connection to the stroma lamellae is called the margin. Grana cores are rich in PSII and LHCII proteins. DGDG contributes to the stacking of the thylakoids in the grana [5] while PG is important for the trimerization as well as the stabilization of LHC complexes [7]. Stroma lamellae and margins are mainly composed of PSI, LHCI as well as around half of the Cyt *b*_6_*f* complex. Margins contain more lipids per protein content than grana stacks and stroma lamellae [8]. According to Duchêne et al. [9], the distribution of the individual glycerolipids is identical in grana stacks, margins and stroma lamellae. 

Lipids also have an important role as integrated components of the photosynthetic reaction centers, and are required for the optimal functioning of the photosystems. In cyanobacteria, absence of PG inhibits electron transfer between Q_A_ and Q_B_ at the acceptor side of PSII [10]. The crystal structure of PSII from cyanobacteria revealed the presence of four molecules of SQDG [11]. In the absence of SQDG, the electron flow from water to tyrosine Z is inhibited, possibly due to conformational changes in PSII [12]. According to Schaller et al. [13], SQDG influences the assembly of PSII core subunits and the antenna complexes. Several lipids were found in the structure of PSI from pea, with one being important for the connection between the Lhc subunit Lhca1 and the PSI subunit PsaF [14]. 

## 3. Dynamic Changes of Supercomplex Formation Required for Cyclic Electron Flow

Depending on the physiological conditions, a higher ATP/NADPH demand is required to fulfil the metabolic demands. Cyclic electron flow allows the generation of a proton gradient without the production of NADPH and serves to provide extra ATP. Alternatively, reduction of O_2_ at PSI, the so-called Mehler reaction or pseudocyclic flow, also leads to an increase in the ΔpH and ATP synthesis. By increasing the ΔpH, cyclic electron flow contributes to the qE component of NPQ and to the regulation of linear electron flow at the level of the Cyt *b_6_f* complex in so-called photosynthetic control. In addition, it may help to increase the volume of the lumen by pumping extra H^+^ into the lumen in the light. Swelling of the lumen facilitates diffusion of plastocyanin and may also allow the relocalization of luminal proteins like, for example, Deg proteases that are involved in the degradation of the D1 protein of PSII [15].

There are two pathways of cyclic electron flow, the first is insensitive to the inhibitor Antimycin A and involving the NDH complex, while the second is Antimycin-A-sensitive and involves the Cyt *b*_6_*f* complex as well as the proton gradient regulation complex PGR5/PGRL1. A protein supercomplex composed of PSI with LHCI, LHCII, Cyt *b*_6_*f* complex, FNR and the integral membrane protein PGRL1 has been isolated from *C. reinhardtii* and suggested to be a prerequisite for cyclic flow [16]. FNR may change its interaction partners and sub-chloroplast location as part of a mechanism to decrease CEF and increase LEF during light adaptation. FNR has been discussed for a long time to be either present in a soluble form in the stroma, or tethered to the thylakoid membrane by interaction with partner proteins. Recently, it has been demonstrated that FNR is always associated with the membrane [17]. This view that state transitions are a required for supercomplex formation was questioned by Takahashi et al. [18] who found the same protein composition of supercomplexes in *C. reinhardtii* wild type, and in a mutant incapable of performing state transitions. According to their data, there is no correlation between state transitions and cyclic electron flow, but rather between the latter and chloroplast redox state (anoxic versus oxic conditions). Furthermore, they reported the presence of PGRL1 together with FNR in high molecular weight fractions that did not contain PSI or Cyt *b*_6_*f* complex. In this context, it is interesting to mention that thioredoxin m forms a complex with PGRL1 [19,20], which itself contains redox-active cysteine residues [21]. The formation of a complex between reduced thioredoxin m and PGRL1 may inhibit cyclic electron flow by preventing the supercomplex formation required for cyclic flow. The work by Takahasi et al. [18], and the reports on the interaction between thioredoxin m and PGRL1 [19,20], indicate that formation of supercomplexes in the stroma lamellae depends on the redox state of the plastoquinone pool, and on the general redox state of the chloroplast.

The example of cyclic electron flow demonstrates that the protein localization is very dynamic, that it is likely under the control of the redox state of the chloroplast and that protein movements within the thylakoid membrane are needed for acclimation responses.

## 4. Protein Movements Involved in Non-Photochemical Quenching

One of the strategies to protect the photosynthetic apparatus from photo-oxidative damage is to dissipate excess light energy as heat at PSII, the so-called NPQ. NPQ is composed of several components that can be distinguished by their relaxation time in the darkness: qE (seconds to minutes), state transitions, qT, (tens of minutes), and photoinhibition, qI (hours). A long persisting quenching related to photoinhibition is called qH. Here, we focus on the recent research progress on dynamic changes in protein–membrane association in NPQ, especially qE, qT (see also the recent review by Johnson and Wientjes [22]), and we will finish by introducing a newly found slowly relaxing quenching: qH.

### 4.1. qE Quenching

qE quenching is the largest component of NPQ in short-term regulations in eukaryotic phototrophs. It occurs within the PSII antenna, the so-called LHCII. LHCII is the most abundant integral membrane protein in chloroplasts and binds 13–15 Chl *a* and Chl *b* molecules [23], 3–4 carotenoids [24] and one phospholipid [25] per monomer. Lumen acidification in the light changes the macro structure of thylakoid membranes affecting the associations between proteins and thylakoid membranes. Different models for the molecular mechanisms of qE in plants and algae have been proposed in the last decades, and they are still a matter of debate. In the following, we describe qE related to (a) the pH-dependent de-epoxidation of violaxanthin to the quencher zeaxanthin, (b) the protein PsbS and (c) to aggregation of LHC proteins. 

(a) One proposed qE mechanism depends on the formation of zeaxanthin as a quencher of excited states of chlorophyll. The xanthophyll cycle enzyme violaxanthin de-epoxidase (VDE), converting violaxanthin to zeaxanthin with the help of ascorbate, is an example of dynamic association of a protein to the thylakoid membranes. It is associated with the membrane depending on the pH in the lumen. VDE is localized at the luminal side of the thylakoid membrane and is attached to the membrane domain containing LHCII and being rich in MGDG when the luminal pH is approximately at 6.5 or less [26,27]. The structure of VDE at pH 7 is very different from its structure at pH 5 [28]. VDE is a monomer at pH 7 and a dimer at pH 5. At pH 5, a hydrophobic patch, likely to be involved in the membrane attachment of VDE, becomes surface exposed [28]. 

(b) The qE-related pH-sensing protein PsbS affects the extent of qE. The protein PsbS does not contain pigments itself but it is involved in zeaxanthin-dependent quenching [29,30]. PsbS has been shown to be associated to PSII, and its protonation state is essential for activating qE. Fan et al. [29] resolved the structure of PsbS, and they discovered structural differences between the PsbS dimers at pH 7.3 and pH 5.0. According to their data, a conformational change at the luminal site may activate PsbS interaction with LHCII and thereby induce the quenching event. Kereiche and coworkers [31] proposed that PsbS controls the macro-organization of the grana membrane. PsbS was not exclusively found in LHCII-PSII supercomplexes, but was also found in PSI fractions [32]. The PsbS content seems to affect both, the grana stacks and the organization of photosynthetic complexes in the stroma lamellae, as demonstrated in freeze-fracture electron micrographs from isolated chloroplasts from wild type, *npq4* (mutant lacking PsbS) and L17 (mutant over-expressing PsbS) [33]. Haniewicz et al. [34] have shown that PsbS is mainly associated with monomeric PSII. This observation suggests that PsbS may play, in addition to its involvement in qE, a role in the PSII repair cycle where dimeric PSII dissociates into monomeric PSII (see below).

(c) According to other models, qE is induced by aggregation of LHCII [35]. Quenching by aggregation can potentially occur in any form of LHCII but more likely it takes place within the major LHCII. LHCII aggregation is caused by the protonation of the antenna at the luminal side. Zeaxanthin is not necessary for the quenching, but may be important to couple LHCII aggregation with a change in ΔpH. The thylakoid membrane becomes thinner and hydrophobic in response to decreases in the lumen pH and the increase in NPQ [36]. Such changes may increase the probability that part of the LHCII detaches from PSII and aggregates. The lipid DGDG plays an important role in the macrostructure of the thylakoid membrane since it contributes to grana stacking [5], and it could favor the contact between LHCII trimers for excess energy dissipation [7]. Ostroumov et al. [37] recently observed a far-red emitting state in isolated aggregated LHCII and in LHCII crystals. This special state could explain how energy is dissipated. According to these authors, qE is derived from a charge transfer reaction involving a chlorophyll pair caused by a conformational change in aggregated LHCII [37]. In another model, qE is based on charge transfer between a chlorophyll-carotenoid pair. This model needs carotenoids such as zeaxanthin or lutein as the essential component for qE [38,39], and may require a closer association of PsbS with qE compared to the other models. In summary, the exact quenching mechanism is still controversial, although it is broadly accepted that qE occurs at the level of LHCII and involves carotenoids and PsbS.

### 4.2. qT: State Transition

State transition is defined as the migration of LHC from PSII (state 1) to PSI (state 2), increasing PSI antenna size and lowering excitation pressure in PSII (for reviews see [40,41,42]). The mechanism of state transition in plants involves phosphorylation of LHCII and detachment of phosphorylated LHCII from PSII. In physiological conditions, it takes place within minutes after a stress, and particularly during fluctuating light [43] or moderate heat stress [44]. State transitions can also be activated artificially by using specific wavelengths that preferentially excite PSII or PSI to change the redox state of the PQ pool (see below) [45]. In higher plants, state transition mobilizes approximately 15–20% of the LHCs. 

The composition in Lhcb proteins of LHCII is complex. Arabidopsis has no fewer than 18 Lhcb proteins [46]; here, only the role of Lhcb1 to Lhcb6 is described. Lhcb 4 to 6 are known to be monomeric and interact directly with the PSII core whereas Lhcb 1 to 3 can form different combination of trimers interacting with monomeric Lhcbs, and these complexes are able to migrate differently during state transition. Three types of trimers are currently defined depending on their interaction with PSII: S-LHCs (strongly), M-LHCs (moderately), and L-LHCs (loosely bound). The latter are known to migrate more easily due to their composition of Lhcb1 and 2 [47]. They are defined as antennas shared between PSI and PSII. It has been shown that L-LHCs constitute almost 50% of the total trimers in plants cultivated in non-saturating light [40]. On the contrary, due to the presence of Lhcb3, M-LHCs are less mobile but their absence affects LHCII reorganization during state transition [48]. S-LHCs seem to be tightly associated with PSII, facilitating light harvesting at the periphery of PSII and ensuring efficient energy capture and transfer to the reaction center [49].

The specific thylakoid-bound transmembrane kinase responsible for state transitions, Stt7, was first discovered *in C. reinhardtii* [50]. Two years later, STATE TRANSITION 7 (STN7) serine/threonine kinase, the homologue of Stt7 in *A. thaliana*, was identified [51]. Activation of the kinase involves both the reduction state of the plastoquinone pool and the Cyt *b*_6_*f* complex. When PSII is preferentially excited, over-reduction of the plastoquinone pool leads to the docking of a plastoquinol molecule to the Qo site of the Cyt *b*_6_*f* complex. This activates the transmembrane kinase that phosphorylates LHCII [52,53,54]. Mutant analysis revealed that the activation of Stt7 depends on its interaction with the stromal side of Cyt *b*_6_*f* complex, whereas its release for LHCII phosphorylation is controlled by plastoquinol occupancy and turnover at the Qo site [53]. In addition, redox regulation via the reduction of a disulfide bridge plays a role in controlling the activity of the Stt7/STN7 kinases [55]. Recently, Wu and coworkers [56] showed that STN7 interacts with the Lumen Thiol Oxidoreductase 1 (LTO1), a transmembrane protein with a luminal thioredoxin domain. This interaction likely helps to keep STN7 in its oxidized active state required for LHCs phosphorylation. These data demonstrate a link between state transitions and the redox state of the chloroplast and especially that of the lumen. For a long time, phosphorylation of LHCIIs was the only known factor required for migration of LHCII. The STATE TRANSITION 8 kinase (STN8), an STN7 isoform, had been shown to be mainly involved in the phosphorylation of PSII core subunits such as D1, D2 and CP43 [57,58,59]. However, recent investigations have demonstrated that STN8 is also secondarily involved in LHCII phosphorylation at the same phosphorylated sites as the STN7 kinase, with a notable preference for Lhcb1 compared to Lhcb2 [60]. This is consistent with STN7 and phosphorylated Lhcb2 being more abundant in stroma lamellae, whereas STN8 and phosphorylated Lhcb1 locate preferentially in the grana core [61]. Recently, Koskela and coworkers [62,63] discovered that phosphorylation of LHCII is not sufficient to induce its migration alone. Plants lacking the acetyltransferase NUCLEAR SHUTTLE INTERACTING (NSI) are locked in state 1, despite phosphorylation of LHCII. NSI acetylates lysins of several proteins including subunits of PSI, PSII and LHCII. Taking together, these findings demonstrate that post-translational modifications that increase the negative charges of the LHC proteins are key elements for state transition. The return to state 1 is initiated by LHCII dephosphorylation mediated by THYLAKOID-ASSOCIATED PHOSPHATASE 38/PROTEIN PHOSPHATASE 1 (TAP38/PPH1) phosphatase [64,65].

How exactly LHCIIs migrate from PSII to PSI along the thylakoid membrane is still unknown. Several hypotheses can be found in the literature: (1) LHCIIs are transferred from grana stacks to stroma lamellae or (2) PSI approaches grana margins where, after a reorganization of the margin domain, an association of the PSI-LHCI-LHCII supercomplex takes place [66,67]. Phosphorylation of Lhcb1 could increase its mobility and the removal of peripheral antenna from PSII complexes. Furthermore, it could permit unstacking of grana [68,69,70]. However, other experiments revealed that phosphorylated Lhcb1 is not included in LHCII trimer binding to PSI [71,72]. Further characterization of a protein discovered 15 years ago gives new insights. CURVATURE THYLAKOID1 (CURT1) B protein, also known as TMP14, was identified by the same phosphoproteomic approaches as STN7. Initial experiments have shown that CURT1B is only localized in PSI fractions, and that it interacts with PsaL, H and O subunits [73]. Furthermore, CURT1B is present in specific regions of grana margins, i.e., the curvature of the grana edges where it is responsible for changes in grana dimensions according to light quality and quantity. CURT1B phosphorylation correlates with LHCII phosphorylation and PSI-LHCII interaction and, according to Trotta and coworkers [74], is key to modulation of thylakoid lateral heterogeneity and/or the decrease in grana diameter.

The interaction between PSI and LHCII was shown to be as strong as the binding of LHCII to PSII [75]. Structural analysis has highlighted that PSI-LHCI-LHCII complex is composed of a unique LHCII trimer, which binds to the PSI core on the opposite side of LHCI. However, it has also been reported that supercomplexes with a second LHCII trimer can bind to PSI to another site involving LHCI [76,77,78,79]. Pan and coworkers [80] recently published a structure showing interaction of PSI-LHCII of maize, involving the PSI subunits PsaO, L, H and potentially PsaK. According to this structure, PsaO, H and particularly PsaL interact directly with phosphorylated Lhcb2. PsaO seems to be involved in light energy transfer from Lhcb2 to PSI core, while PsaK could function similar for Lhcb1.

### 4.3. qI: Photoinhibition

Photoinhibition is characterized by a loss of maximum chlorophyll fluorescence and a loss of water-splitting activity in PSII. Photoinhibition is a process that occurs when the photo-oxidative damage to PSII is greater than its repair [81]. This phenomenon takes place when plants are facing stressful conditions, such as higher light intensities than required for photosynthesis. In the most drastic cases, it is characterized by a sharp decrease in the photosynthetic capacity of the plant due to loss of linear electron transport. The D1 protein of PSII is considered in the literature to be the principal protein damaged under photoinhibitory conditions [82,83,84]. It is known for having a very rapid turnover in the order of 20 min [85,86], thanks to a very efficient repair system. Under photoinhibitiory conditions, the acceptor side of the PSII is over-reduced, which results in the production of reactive oxygen species and, in particular, singlet oxygen that is responsible for the damage of D1 [87]. In addition, photoinhibition can also take place at the donor side of PSII inhibiting the [Mn_4_CaO_5_] cluster, thereby limiting electron donation to Tyr_Z_^+^ and P680^+^ [88].

The movement of proteins within the thylakoid membranes and the association of proteins with the membrane/photosystems are important for not only qE and qT, but also for the PSII repair process. Active PSII dimers are located in grana stacks, whereas the PSII repair cycle occurs in grana margins. First, damaged PSII centers are phosphorylated by STN8 leading to the disassembly of LHCII-PSII complexes and PSII dimers [57,58,89]. However, a study of the *stn8* mutant highlighted that the repair is not completely dependent on phosphorylation [57] but that the later facilitates lateral mobility and, therefore, the transport of damaged PSII from the grana stacks to the margins [66]. A study of fluorescence recovery in a confocal laser-scanning microscope suggested that a limited population of chlorophyll-containing proteins can move within a 10-min time scale [90]. The population of mobile proteins increased after PSII photoinhibition in wild type but not in the *stn8* mutant of *A. thaliana*. This suggests that migration of damaged PSII depends on phosphorylation of certain subunits. In the grana margins, photoinhibited PSII is dephosphorylated again in order to allow partial disassembly of PSII monomers and to trigger degradation [91,92]. Upon arrival at grana margins, only the intermediate PSII complex RC47 remains, which is recognized by the protease FtsH [93,94]. FtsH is a complex of metalloproteases anchored to the thylakoid membrane, and it is important for the formation of thylakoid membranes during chloroplast development as well as in their maintenance [95,96]. In addition to FtsH, stromal and lumenal Deg proteases act in a cooperative manner, allowing the endopeptidase cleavage of the D1 protein. It is assumed that the action of both stromal and lumenal Deg proteases makes it possible to increase the number of D1 recognition sites for FtsH, and to facilitate the dissociation of the water-oxidizing complex [97,98,99].

A new quenching component, qH, active during photoinhibition and completely independent from the other mechanisms cited so far, has recently been reported [100,101]. qH has been described as a slowly relaxing quenching with photoprotective action which acts in concert with other quenching mechanisms involved in qI. The site of qH has been localized in the peripheral antenna of PSII. Until now, three proteins have been identified to be involved in qH: LCNP, a plastid lipocalin protein, is necessary for qH activation while for the relaxation of qH, two proteins were identified so far: SOQ1 for SUPPRESSOR OF QUENCHING 1 and the protein RELAXATION OF QH1 (ROQH1). LCNP-mediated modification of a thylakoid membrane lipid was proposed to change the conformation of LHCII and thus create a quenching site [100]. SOQ1 is a transmembrane protein that constitutively inhibits qH. LCNP is located in the thylakoid lumen and may interact with the SOQ1 domains responsible for regulating qH. ROQH1 is an atypical short-chain dehydrogenase-reductase and functions as a qH-relaxation factor. It may either directly remove the quenching conformation of the antenna or modify the hydrophobic environment at the LHCII. An *A. thaliana soq1roqh1* double mutant shows constitutive qH [101].

## 5. Proteins with Dynamic Changes in Oligomerization State and Localization at the Thylakoid Membrane

In the following, we introduce two proteins (PTOX, VIPP) that are related to the response to high light stress. These proteins are activated by perturbation of the redox poise of the photosynthetic electron transport chain. They permit a protection of the photosynthetic electron transport chain against over reduction (PTOX), or play a role in the repair of damaged PSII (VIPP). The common point of these proteins is changes in their oligomerization state and their association with the thylakoid membrane, depending on the physiological condition.

### 5.1. Plastid Terminal Oxidase (PTOX)

The plastid terminal oxidase (PTOX), a plastohydroquinone:oxygen oxidoreductase of the thylakoid membranes, contains a catalytic center similar to that of the alternative oxidases of the mitochondria (AOXs), and the two enzymes share structural similarities [102]. PTOX is involved in carotenoid metabolism, it protects the photosynthetic electron transport chain against over-reduction, it participates in chlororespiration, and it may be important for setting the redox poise for cyclic electron transport. PTOX has been localized in the region of the stroma lamellae in spinach chloroplasts of non-stressed leaves [103]. It was recently demonstrated that PTOX can relocalize within the membrane [104], or can even reversibly associate to the membrane depending on the proton motive force [105]. Stepien and Johnson [104] showed that PTOX overexpressed in the halophile *Eutrema salsugineum* moved from the stroma lamellae to the grana stacks upon exposure of the plants to salt stress. Using confocal fluorescence microscopy, Bolte and coworkers [105] investigated PTOX localization in *A. thaliana* overexpressing green fluorescent protein (GFP)-PTOX. They observed a relatively homogeneous distribution of the GFP fluorescence over the whole chloroplasts when leaves had been pre-illuminated with high light intensities, a condition generating a high proton motive force. In contrast, when leaves were treated with uncouplers, or had been adapted to the dark, the GFP fluorescence was localized in spots. The spots were interpreted as PTOX oligomers. These results show that PTOX localization within the chloroplast is highly dynamic and depends on the pH in the stroma, and/or on the ion concentration. Membrane association with the grana lamellae is required to allow PTOX the access to its lipophilic substrate plastoquinol. In vitro, purified PTOX associated at slightly alkaline pH-value to liposomes composed of a lipid mixture similar to thylakoid membranes. Enzymes of the carotenoid biosynthesis pathway may behave in a similar manner like PTOX. As mentioned above, at low pH in the lumen, VDE changes its conformation, dimerizes and attaches to the membrane, thereby gaining access to the lipophilic violaxanthin. Furthermore, the phytoene desaturase that requires PTOX activity for the regeneration of its electron acceptor, plastoquinone, seems also to change its oligomeric state and thereby its activity. Gemmecker et al. [106] reported that phytoene desaturase exists in vitro in rings or stacks. The rings were composed of tetramers, which assembled into highly ordered stacked tubular structures of a length between 15–30 nm. The oligomeric stacks were soluble, while the form attached to liposomes was assigned to a single tetrameric ring. It is likely that the same distribution for phytoene desaturase is found inside the chloroplasts: a soluble form organized in stacks in the stroma and the smaller tetramers attached to the thylakoid membrane, allowing access to its substrate phytoene. Similar changes in the oligomerization state as described for the phytoene desaturase may also take place for PTOX. The catalytically functional unit of PTOX consists most likely of a dimer as it is the case for AOX [107]. Recombinant fusion protein maltose-binding protein-PTOX (MBP-PTOX) of rice tends to form different oligomeric states depending on the detergent used. In the presence of *n*-octyl *β*-D-glucopyranoside, MBP-PTOX was found mainly as a tetramer [108], while it was mainly in a dimeric form in the presence of *β*-dodecyl-maltoside [105]. Angiosperms contain one PTOX isoform, while other phylae mostly contain two isoforms; a plant-type PTOX and an algal-type PTOX [109]. The algal-type PTOX has a longer N-terminal extension that probably allows a stronger anchoring of this protein to the membrane. A dynamic association of PTOX with the thylakoid membrane may be important to permit PTOX activity only under conditions of a highly reduced electron transport chain, thereby preventing competition between PTOX and Cyt *b*_6_*f* complex for platoquinol under optimal conditions for photosynthesis. The same holds for the VDE, permitting the formation of antheraxanthin and zeaxanthin only under certain physiological conditions. The activity of other carotenoid biosynthesis enzymes, such as phytoene desaturase, may be regulated in a similar way.

### 5.2. Vesicle-Inducing Proteins in Plastids (VIPP) 

Another example of a stroma protein that changes its oligomerization state, and its membrane association, is the vesicle-inducing protein in plastids (VIPP). Plants contain only one isoform, VIPP1, while chlorophyceae have a paralog of VIPP1, called VIPP2. In chloroplasts, VIPP1 has been found at different location within the chloroplast: in a soluble form in the stroma, and in a membrane-associated form at the chloroplast envelope and at the thylakoids. In vitro, VIPP1 and VIPP2 form large oligomeric rod-like structures [110,111]. Fluorescence microscopy in vivo revealed high oligomerization states of VIPP1 as punctae, indicating that VIPP1 may assemble into higher-order oligomers not only in vitro but also in vivo [112,113]. The chloroplast chaperone system, with HSP70B-CDJ2-CGE1 being interaction partners of VIPP, plays a role in controlling the oligomerization state of VIPP [110]. A role in membrane vesicle traffic and in the biogenesis of the lipid part of thylakoid membranes had been originally attributed to VIPP1 [114] while it was later shown to be also involved in high light stress response [112,113,115]. In cyanobacteria, VIPP1 punctae dynamically form at or close to highly curved thylakoid membranes. At high light intensities, the membrane-bound form of VIPP1 was favored compared to lower light intensities when more of the cytosolic form was present [112,113]. In *C. reinhardtii*, both VIPP1 and VIPP2 accumulate under photoinhibitory conditions and likely play a role in the PSII repair cycle [115]. A certain level of VIPP1 proteins was present in all light conditions, while VIPP2 was only detectable after high light stress [116]. In *C. reinhardtii*, VIPP1 knock-down lines showed defects in PSII biogenesis and in repair of photoinhibited PSII [115]. These results indicate that VIPP1 has an important role in creating membrane domains that favor disassembly and repair of photoinhibited PSII.

## 6. Conclusions and Perspectives

In the present review, we give examples that show the physiological importance and the complexity of dynamic changes of protein associations to and within the thylakoid membrane. The photosynthetic electron transport components (PSII, PSI, LHCII, etc.) are heterogeneously distributed in the thylakoid membrane. The protein composition of supercomplexes changes depending on the physiological situation. Protein movements play a role in short-term regulatory mechanisms that protect the photosynthetic apparatus against photo-oxidative damage. Protein–membrane associations are required for the PSII damage–repair cycle and for the biogenesis of the photosynthetic apparatus in general. Dynamic changes of membrane association of VDE, phytoene desaturase and PTOX allow a regulation of their enzymatic activity by allowing access to their lipophilic substrates. In addition, local differences in pH and in ion concentrations are likely, creating domains which may favor membrane association of proteins. At present, it is unknown to which degree such local domains with differences in proton or ion concentration are separated or shared, and how much (how fast) changes in these heterogeneities are associated with the regulation of photosynthetic electron transport.

Many questions remain open like the effects of ions and pH on the relocalization of proteins, factors that determine the oligomerization state of proteins, associated lipids or protein partners that allow changes in oligomerization states and in the association of proteins to the membrane. Recent development of super resolution microscopy, and fluorescence lifetime imaging on plants expressing fluorescent proteins, will allow better characterization of dynamic protein localizations and structural rearrangements of the thylakoid membrane upon exposure of plants to changes in the environment. 

## Figures and Tables

**Figure 1 cells-10-01216-f001:**
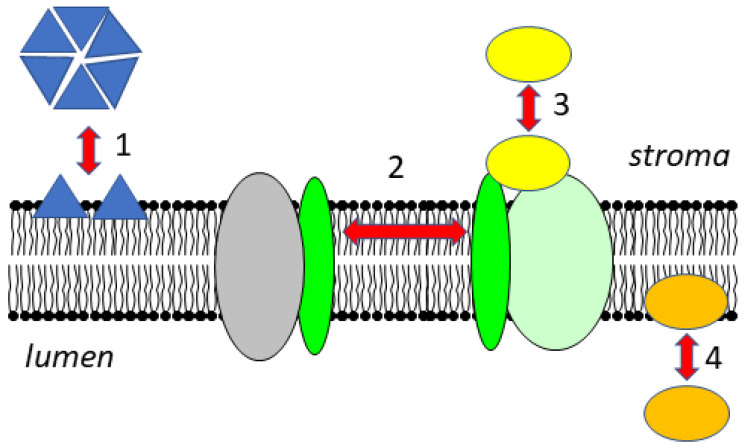
Dynamic changes of the association of proteins with the thylakoid membrane. In the stroma, proteins can be present in soluble forms or as peripheral membrane proteins, and their oligomerization state can change upon membrane association (**1**). Transmembrane proteins can move laterally within the membrane and are associated with different partner proteins (**2**). Soluble proteins can attach to protein complexes in the membrane (**3**). In the lumen, soluble proteins can attach to the membrane as a function of pH (**4**).

**Figure 2 cells-10-01216-f002:**
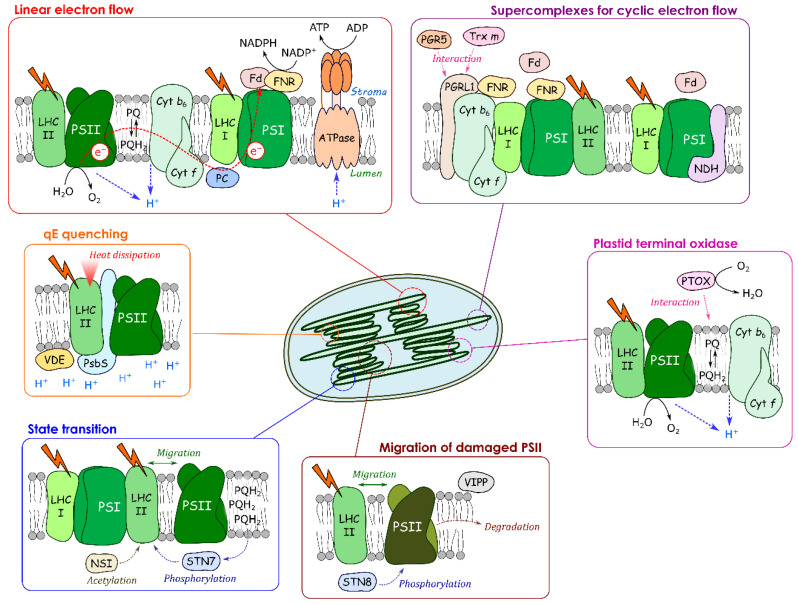
Dynamic changes of protein–membrane association for regulating photosynthetic electron transport. Photosynthetic linear electron flow produces NADPH and ATP for CO_2_ assimilation in the grana margin. In the grana, heat dissipation at the light-harvesting complex (LHCII), the so-called qE quenching, occurs, and the damaged PSII migrates and is digested; In the grana margin, STN7 and NSI stimulate the state transition. PTOX changes its localization in response to stroma pH to maintain the PQ redox poise. A part of PSI forms supercomplexes with PGRL1, FNR and NDH in the stroma lamellae.

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
