# Peer review of "Dynamic Changes in Protein-Membrane Association for Regulating Photosynthetic Electron Transport"

_cells, 2021, doi:10.3390/cells10051216_

Round 1

Reviewer 1 Report

The manuscript by Messant et al. is a review of a deep involvement of protein-membrane association in photosynthetic electron transport regulation, as clearly shown in its title. The manuscript is totally well organized with covering latest important findings as well as fundamental understandings, although the relationship between lipid composition of chloroplast membranes mentioned in the first place of the manuscript and NPQ mechanisms and protein-membrane associations described in the later parts seems somewhat unclear, presumably because of lack of sufficient evidence to connect them.

I have only one comment for revision, that indicates a missing of references for the second paragraph in page 4, lines 14-24, particularly for the grana core composition rich in PSII-LHCII and DGDG and the association of PG with DGDG for trimerization and stabilization of LHCs.

Author Response

We have added two citations: Demé et al (2014) for DGDG and Liu et al (2004) for the role of PG in trimerization and stabilization of LHCII.

Reviewer 2 Report

This review reports on dynamic changes  in the thylakoid membranes following changes in light conditions.  It starts with a  discussion on the lipid composition of thylakoid membranes and on linear and cyclic electron flow. This part  provides a balanced view. The remaining part of this review is mostly focused on  NPQ-qE, state transitions and the PSII repair cycle  as well as on changes in oligomeric states and association to  thylakoid membranes of PTOX,  VIPP1 and WHIRLY1. Moreover  the potential role of  WHIRLY in retrograde signaling is briefly outlined. One problem with this reviewis that  it covers too much and in some parts in a superficial and incomplete way. In particular, there are several flaws  in the section on state transitions. Some of references are incorrect and some of the results in this area are ignored (for details see below)

  1. The review on state transitions of J. Allen (1992) is  rather old and several more recent reviews have been published  on this topic.
  2. The original reports on the Stt7/STN7 kinase are not mentioned (Depège et al. 2003; Bellafiore et al. 2005) . Certainly ref. 41 is not the original paper describing Stt7/STN7 as claimed by the authors
  3. Contrary to what the authors state the Stt/STN7 kinase is activated both through the lumen and stromal side of the thylakoids (see Shapiguzov et al. 2016; Wu et al. Plant Phys. (2021) DOI: 1093/plphys/kiab091
  4. The importance of the PSI_LHCI-LHCII complex for state transitions has been seriously questioned by Takahashi et al 2013. This important publication should be at least discussed.
  5. The crystal structure of PsbS determined by Pan et al (2015) should be mentioned and discussed.
  6. The first publication on the structural basis for the pH-dependent xanthophyll cycle enzyme VDE of Arnoux et al (2009) should be mentioned.
  7. The section on retrograde signaling is too sketchy. This part should be either further developed or removed.
  8. Some sentences need to be rephrased; e.g. bottom of p. 6 “… which is likely to increase the changes that a part of the LHCII detaches from PSII and ”

Author Response

We have largely revised the section on state transitions.

To 1. We have added two more recent reviews (Rochaix, 2014; Dumas et al., 2016).  (l. 233)

To 2. The original reports by Depège et al. 2003 and Bellafiore et al. 2005 are now cited (line 255-257).

To 3. We have corrected this point. We have added the following sentences “In addition, redox regulation via the reduction of a disulfide bridge plays a role in controlling the activity the Stt7/STN7 kinases (Shapiguzov et al., 2016). Recently, Wu and coworkers (2021) showed that STN7 interacts with the Lumen Thiol Oxidoreductase 1 (LTO1), a transmembrane protein with a luminal thioredoxin domain. This interaction likely helps to keep STN7 in its oxidized active state required for LHCs phosphorylation.” (l. 264-270)

To 4. In the revised version we discuss the work by Takahashi in the section on cyclic electron transport (l. 148-153).

To 5. The crystal structure of PsbS by Fan et al is now cited (l. 198)

To 6. The work by Arnoux and colleagues is now described (l. 192-194).

To 7. We agree with the reviewer and decided to remove the part on WHIRLY and on retrograde signalling both from the text and from Figure 2.

To 8. Thomas Roach, a native speaker, kindly corrected our English.

Round 2

Reviewer 2 Report

The authors have satisfactorily addressed the points I raise in the first review. Only a few corrections need to be done:

Fig.2 legend, line4  Change “PSII is migrated and digested” to “PSII migrates and is digested”

                        line 5 Correct the  sentence to “PTOX changes its localization in response to the stromal pH to maintain the PQ redox poise”

Line 449/450 correct to “… important role in creating membrane …”

Author Response

We have corrected all sentences as suggested by the reviewer.